# Evaluating the Impact of Sight Distance and Geometric Alignment on Driver Performance in Freeway Exits Diverging Area Based on Simulated Driving Data

Xizhen Zhou [1] , Binghong Pan [1,*] and Yang Shao [2]

1   Highway School, Chang'an University, Xi'an 710064, China; zhouxizhen@chd.edu.cn
2   School of Modern Posts (Logistics School) & Institute of Posts, Xi'an University of Posts & Telecommunications, Xi'an 710061, China; shaoyang2020@xupt.edu.cn
*   Correspondence: panbh@chd.edu.cn

**Abstract:** The decision sight distance (DSD) at freeway exits is a major factor affecting traffic safety. Based on the Hechizhai Interchange in Xi'an City (Shaanxi Province, China), this paper designs a simulation experiment. Through a simulator study and a questionnaire survey, this paper discusses the impact of the DSD, 1.25 times the stopping sight distance (SSD) and a circular curve deflection on a driver's driving state (including steering wheel angle rate and steering wheel angle frequency domain). Thirty volunteers participated in this research. The result shows that (1) it is safer to drive on an exit that meets DSD. (2) If it only meets the 1.25 times the SSD requirement, the overloaded driving tasks and operation would be more likely to cause crashes. The driving state of the driver on the right circular curve is obviously better than that on the left circular curve, because changing lanes to the right on the left circular curve does not meet the driver's expectations. (3) Left and right circular curve should be treated differently in the driving area and the constant sight distance requirements should not be applied. (4) The left circular curve should be more stringent to ensure driving safety.

**Keywords:** freeway; diverging area; freeway design; sight distance; driving safety

## 1. Introduction

Freeway exit areas have the highest crash rate in the whole freeway system [1] and it is a special place [2]. According to American survey statistics research, crashes in the diverging area account for 44% of the total interchange area [3]. Some scholars [4] classified 1150 collision crashes on interchanges in Northern Virginia and found that 48% of the crashes occurred at the ramp exit and 36% occurred at the ramp entrance. A total of 2400 truck crashes occurred on the freeway in the American state of Colorado in the three years from 1993 to 1995, of which about 30% occurred at the ramp junction [5].

Driving behavior is closely related to road safety [6–10] and safety at a diverging area is influenced by many factors [11,12]. The traffic organization and traffic flow operational state in a diverging area are complex. The traffic behaviors of straight, deceleration, interleaving and lane changing at an exit ramp have a significant impact on traffic safety at an exit [13]. In order to improve safety at this section, many studies have been conducted to find the cause of crashes. The research mainly focuses on the relationship between acceleration and deceleration lanes [14–19] and crashes at an entrance and exit [4,20–24], as well as the relationship between exit types and crashes [2,24–26]. Some scholars also analyzed the setting of exit signs and pavement markings [27,28] and found that crashes were closely connected to sight distance, shoulder width and pavement guidance facilities. Moreover, crashes in a main line were more frequent than those on a ramp [29].

Although some scholars have proposed that the number of crashes on the main line of a diverging area is higher than that on the ramp, not much attention has been paid to this.

There is no relevant literature on the control factors of the main line geometric alignment at an exit and ignores the importance of DSD in ensuring diverging area safety. When driving at an exit ramp with a circular curve, a driver's sight maybe obstructed by the central median, the right guardrail and so on. This may cause DSD insufficiency, then the driver may miss the best time to diverge from the main line. However, after missing the best opportunity, some drivers still forcibly change lanes to driver away, which can easily cause traffic crashes. At the Linhai North Interchange on the Shenhai Freeway in China, 83 traffic crashes occurred during the study period [30]. Among them, 45 crashes were caused by drivers having difficulty discovering an exit. In addition, DSD is largely affected by the radius of the main line circle curve.

At present, there are some studies on the circular curve radius and DSD, but almost all of them ignore the influence of the circular curve deflection on DSD. In 1979, DSD was associated with specific road types, design speeds, traffic operating conditions and could be calculated by establishing a hazard avoidance process model [31]. In this way, the relationship between DSD and road geometric alignment is balanced. Most of these subsequent studies focus on the parameter modification in the DSD and SSD calculation models [32–38]. A Policy on Geometric Design of Freeways and Streets (the Green Book) [39] indicated that the sight distance on a freeway preceding the approach nose of an exit ramp should exceed the minimum SSD for the through traffic design speed, desirably by 25 percent or more. However, the problem of distinguishing the deflection in calculating the radius of circular curve is ignored. Since the deflection of the circular curve is different, the reasons that hinder the driver's sight are also different; therefore, it is unknown whether there is a safety hazard in the current equal treatment. Since it is pointed out that freeway exits needs to meet the DSD, it is debatable whether the 1.25 times the SSD requirement proposed at the same time is safe.

As the discussion above indicates, currently the most discussed relationships are those between the various components of the ramp in a diverging area and the crash. There are few studies on the influence of the main line geometry alignment and sight distance on driving safety. This paper will discuss the influence of sight distance (DSD and 1.25 times the SSD) and circular curve deflection (left circular curve and right circular curve) on driving behavior using driving data, in order to improve the safety of freeway exits. Based on the environmental conditions of the Hechizhai Interchange in Xi'an (Shaanxi Province, China), this article designs a simulation experiment for finding 36 actual exits where meeting the requirements is unrealistic. In addition, this experiment is carried out in a simulator, which can minimize the interference of external factors and make the driving data extracted in this study more valuable. The rest of this article is organized as follows: Section 2 introduces the calculation models for different circular curve radius. Section 3 introduces the field experiment in detail. Section 4 introduces data collection and analysis. Section 5 provides an analysis of the results obtained. Finally, the results of this research are summarized in Section 6.

## 2. Calculation Method

### 2.1. Concept Load

SSD is the length of the roadway ahead that is visible to the driver and it should be sufficiently long to enable a vehicle traveling at or near the design speed to stop before reaching a stationary object in its path [37]. DSD is the distance needed for a driver to detect an unexpected or otherwise difficult-to-perceive information source or condition in a roadway environment that may be visually cluttered, recognize the condition or its potential threat, select an appropriate speed and path and initiate and complete a complex maneuver [37,38]. DSD is designed to provide sufficient decision-making time for a driver. At present, there is no relevant research that has determined the recognition object of the DSD. Due to an opportunity to change lanes provided by the DSD at a diverging area, it should at least be ensured that the driver in the second lane (the lane adjacent to the outermost lane) can identify the variation of the main line within a safe distance, to ensure

that the driver has enough time to obtain the guide sign information, confirm the exit and drive into the deceleration lane in time.

Therefore, in order to fully ensure the safety in a diverging area, the endpoint of the DSD should be placed at the starting point of the transition of the deceleration lane (Figure 1), which allows drivers to identify an exit before entering the deceleration lane and then fully ensure the traffic's safety in a diverging area.

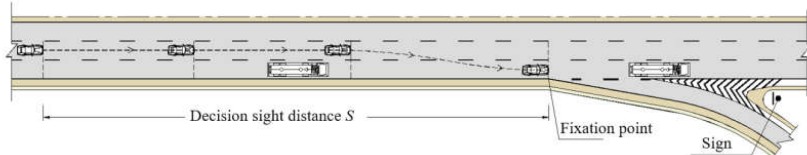

**Figure 1.** DSD of a diverging area.

### 2.2. Calculation of the Radius
### 2.2.1. Left Circular Curve

When the alignment of the main line in a diverging area is a left circular curve, we assume that the vehicle leaving the main line is driving in the second lane. At this time, anti-dazzle facilities and guardrails in the median strip may block the sight of drivers without a sufficient circular curve radius of the main line. As a result, the driver cannot identify the exit within a short period because of the insufficient DSD, which affects the driver's judgment and operation.

A driver's view trajectory is presented in Figure 2. The "SP" in Figure 2 is the driver's viewpoint position and the "TP" is the driver's fixation point position. According to this, the calculation model of the minimum radius of a left circular curve of the main line in a diverging area can be simplified to the geometric model shown in Figure 2.

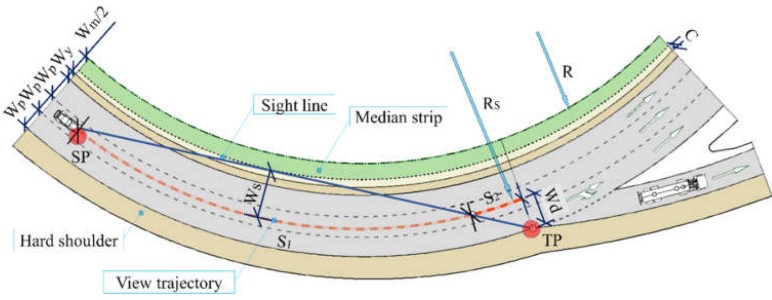

**Figure 2.** A schematic diagram for the minimum radius of a left circular curve of the main line.

According to the geometric relationship shown in Figure 2, the left circular curve radius of the main line can be formulated using Equation (1), as follows:

$$\begin{cases} W_S = R_S\left(1 - \cos\frac{S_1}{2R_s}\right) \\ W_S + W_d = (R_S + W_d)\left(1 - \cos\frac{2S - S_1}{2R_S}\right) \\ W_S = W_y + (n-2)W_p + D_S + C \\ W_d = 2W_p - D_S \\ R = R_S - W_S - \frac{W_m}{2} + C \end{cases} \tag{1}$$

where $W_s$ is the lateral clear distance of the view trajectory (m); $W_d$ is the transverse distance between the TP and view trajectory (m); $R_s$ is the radius of the view trajectory (m); $S$ is DSD (m) and $S = S_1 + S_2$; $S_1$ is the arc length from the intersection of the view trajectory and the line of sight to the viewpoint (m); $S_2$ is the difference between DSD and $S_1$ (m); $W_y$ is marginal strip width of the left side of the main line (m); $n$ is the number of one-way freeway lanes (m); $W_p$ is the width of the lane (m); $D_s$ is the distance between the viewpoint

and the left edge line of the lane (m); $W_m$ is the width of the median strip (m); and $C$ is the lateral clearance of the median strip (m).

### 2.2.2. Right Circular Curve

When the alignment of the main line in a diverging area is a right circular curve, guardrails and obstacles on the right roadside may block the sight of drivers without a sufficient circular curve radius of the main line.

A driver's view trajectory is shown in Figure 3. The "SP" in Figure 3 is the driver's viewpoint position and the "TP" is the driver's fixation point position. According to this, the calculation model of the minimum radius of a right circular curve of the main line in a diverging area can be simplified to the geometric model shown in Figure 3.

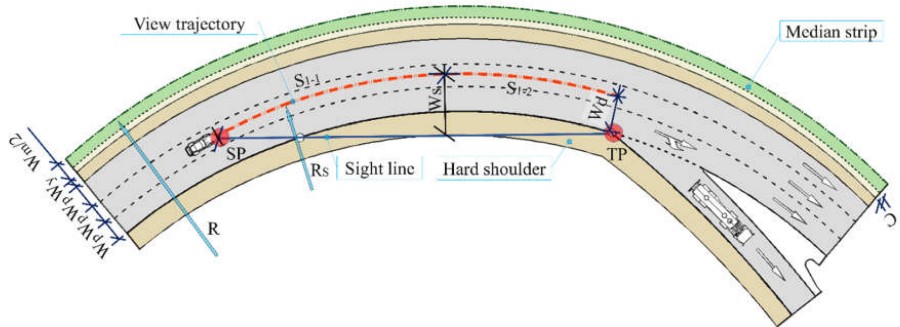

**Figure 3.** A schematic diagram for the minimum radius of a right circular curve.

According to the geometric relationship shown in Figure 3, the right circular curve radius of the main line can be formulated using Equation (2), as follows:

$$
\begin{cases}
S_{1-1} + S_{1-2} = S \\
W_S = R_S \left( 1 - \cos \frac{S_{1-1}}{R_s} \right) \\
W_S - W_d = (R_S - W_d) \left( 1 - \cos \frac{S_{1-2}}{R_s} \right) \\
W_S = 2W_p - D_S + W_h \\
W_d = 2W_p - D_S \\
R = R_S + D_s + (n-2)W_p + W_y + \frac{W_m}{2}
\end{cases}
\tag{2}
$$

where $W_h$ is the hard shoulder width on the right side of the main line (m); $S_{1-1}$ is the arc length from the intersection of the view trajectory and the line of sight to the viewpoint (m); $S_{1-2}$ is the difference between DSD and $S_{1-1}$ (m); and the meaning of the other symbols is the same as before.

### 2.3. *The Position of Viewpoint*

The viewpoint position refers to the distance between a driver's eyes and the left edge of the lane. As there are great differences in the existing research values, this paper uses the method of field investigation and an analysis of vehicles' position distribution on the lane and then calculates the viewpoint position to study a driver's viewpoint position of the second lane of the main line in a diverging area. The investigation sites are Hechizhai Interchange in Xi'an City (Shaanxi Province, China), where the main line has a right circle curve, and Qujiang Interchange, where the main line has a left circle curve. An Unmanned Aerial Vehicle (UAV) was utilized to capture the following images (Figure 4a). The survey was conducted on 19 September 2020 and 20 September 2020. During the survey, the weather was quite clear and windless, and the traffic was free flowing.

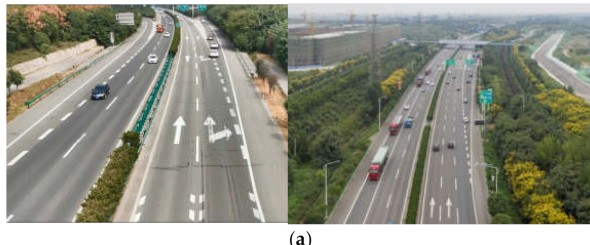 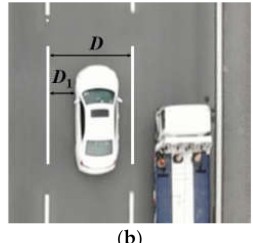

(**a**)                                       (**b**)

**Figure 4.** Interchange diverging area: (**a**) Alignment of diverging area; (**b**) Methods of data extraction.

During the process, ensure that the UAV is located directly above the second lane of the main line of the diverging area and the section is 1 km away from the exit. The UAV flies at an altitude of 800 m. It is known that the width of the lane on this road is 3.75 m. Measure the width of the lane on the map (the value of "$D$" in the Figure 4b) and the distance between the edge of the vehicle and the left edge of the lane (the value of "$D_1$" in the Figure 4b). After scaling, measure the actual distance between the edge of the vehicle and the left edge of the lane (Figure 5 and Equation (3) are shown). Use SPSS software to carry out a K–S normality test on the sample data (Table 1).

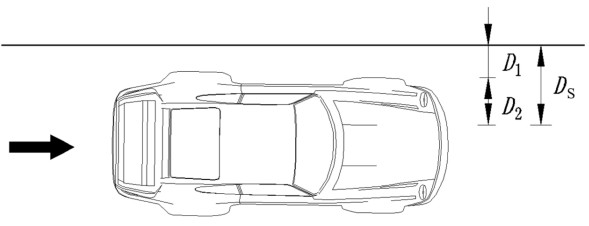

**Figure 5.** Driver's viewpoint position.

**Table 1.** K-S test results of sample dates.

| Vehicle | Type | Small Cars | Medium Cars | Large Cars |
|---|---|---|---|---|
| $D_2$/m | Weight/t | <4.5 | 4.5~12 | >12 |
|  | Sample quantity/unit | 134 | 101 | 117 |
|  | $p$-value | 0.061 | 0.124 | 0.100 |
|  | Sample average $D_2$/m | 0.503 | 0.546 | 0.600 |
|  | $D_2$/m | 0.50 | 0.55 | 0.60 |
| The second lane (Right circular curve) | Sample quantity/unit | 213 | 33 | 26 |
|  | Asymptotic Significance) (2-sided) | 0.200 | 0.200 | 0.200 |
|  | Sample average $D_1$/m | 1.034 | 0.998 | 0.610 |
|  | $Ds$/m | 1.534 | 1.548 | 1.210 |
| The second lane (Left circular curve) | Sample quantity/unit | 223 | 24 | 26 |
|  | Asymptotic Significance) (2-sided) | 0.162 | 0.122 | 0.075 |
|  | Sample average $D_1$/m | 0.778 | 0.543 | 0.513 |
|  | $Ds$/m | 1.278 | 1.093 | 1.113 |

The driver's viewpoint position $D_s$ (Figure 5) is calculated as follows:

$$D_s = D_1 + D_2 \tag{3}$$

where $D_1$ is the distance from the edge of the left vehicle body to the edge of the left lane (m); $D_2$ is the distance from the center of the driver's seat to the edge of the left vehicle body (m); and $D_s$ is the driver's viewpoint position (m).



Considering the purpose of the survey and FHWA's classification of vehicle types [39], this paper divides vehicles into the following three categories: small cars, medium cars and large cars according to vehicle weight (Table 1) and conducted a survey on the $D_2$ (Table 1).

It can be seen from the test results that the double-sided values of large, medium and small cars are all greater than 0.05. The sample data obey the normal distribution, and the sample's average is the expected value. At the same time, the investigation found that the difference in the distance between the center of the driver's seat and the edge of the left vehicle body in different vehicle types is small and the change of the body width of different vehicles is mainly reflected in the change of the distance between the two driver's seats.

The statistics show that, in the case of a right circular curve, the viewpoint values of the small, medium and large cars in the second lane are, respectively, 1.534, 1.548 and 1.210 m. In the case of a left circular curve, the corresponding values are 1.278, 1.093 and 1.113 m. Considering the influence of the viewpoint on the calculation results and the most unfavorable situation, in the following calculations, the viewpoint value of the right circular curve is calculated as 1.6 m and the left circular curve is calculated as 1.0 m.

## 3. Materials and Methods

### 3.1. Participants

According to China Industry Information Network, in 2020, male and female drivers account for 67 and 33% of the total number, respectively. Drivers aged 26–50, 51–60 and 18–26 account for 72, 13 and 15% respectively. At the same time, many studies have proved that, subject to the constraints of experimental conditions, small samples can also be used in driver characteristic tests. Usually, the number of samples should be greater than 6 to be effective [40–44]. In order to better approximate the actual driver age and gender distribution, 30 participants were recruited for the experiment, including 21 male drivers and 9 female drivers. The age distribution ranged from 24 to 51 years old (Mean = 28.9; Standard Deviation (SD) = 5.99), with different driving experiences (Mean = 5.16; SD = 2.166). The test required all of the drivers to have good driving manners, good vision after correction, good health, sufficient rest the day before the test and no intake of stimulating food, such as alcohol and caffeine.

### 3.2. Driving Simulator

The Forum8 Driving Simulator was used in the experiment and this driving simulator has been widely used in research [45,46]. This simulator simulates driving by combining three-dimensional electronic information and, through a combination of virtual reality technology and a cockpit, the experimenter can realistically simulate driving in a three-dimensional scene. The visual system consists of three liquid displays providing a 120 degrees forward field of view.

### 3.3. Driving Scenarios and Environment

The simulation scene is based on Hechizhai Interchange, which is located in Xi'an City (Shaanxi Province, China). The main line is a three-lane road in each direction. The lane width ($W_p$) is 3.75 m, the right shoulder width ($W_h$) is 3 m, the median strip width ($W_m$) is 3 m, the marginal strip width ($W_y$) is 0.75 m and the value of C is 0.25 m. The SSD and DSD are calculated based on the method in the Green Book [37]. According to formula (1,2), the ultimate radius R of the circular curve in the main line satisfying the sight distance is obtained (Table 2). The calculation result is rounded to the nearest 5 m.

**Table 2.** The minimum radius of the main line circular curve that meets the criterion for sight distance.

| Deflection | Design Speed (km/h) | DSD (m) | 1.25*SSD (m) | R Satisfying DSD (m) | R Satisfying 1.25*SSD (m) |
|---|---|---|---|---|---|
| Right | 50 | 195 | 81 | 860 | 155 |
| | 60 | 235 | 106 | 1245 | 260 |
| | 70 | 275 | 131 | 1700 | 395 |
| | 80 | 315 | 163 | 2230 | 605 |
| | 90 | 360 | 200 | 2910 | 905 |
| | 100 | 400 | 231 | 3590 | 1205 |
| | 110 | 430 | 275 | 4150 | 1710 |
| | 120 | 470 | 313 | 4955 | 2210 |
| | 130 | 510 | 356 | 5835 | 2855 |
| Left | 50 | 195 | 81 | 520 | 90 |
| | 60 | 235 | 106 | 755 | 160 |
| | 70 | 275 | 131 | 1040 | 240 |
| | 80 | 315 | 163 | 1365 | 340 |
| | 90 | 360 | 200 | 1780 | 570 |
| | 100 | 400 | 231 | 2120 | 760 |
| | 110 | 430 | 275 | 2545 | 1080 |
| | 120 | 470 | 313 | 3040 | 1400 |
| | 130 | 510 | 356 | 3580 | 1815 |

*3.4. Experimental Design*

According to the 36 situations in the table above, 36 different exits models are established. As shown in Figure 6, the geometric alignment of the main line is connected by the straight line, moderate curve and circular curve. The length of the straight line is 1 km, and the length of the other constituent units is the value required by the corresponding speed in the Green Book.

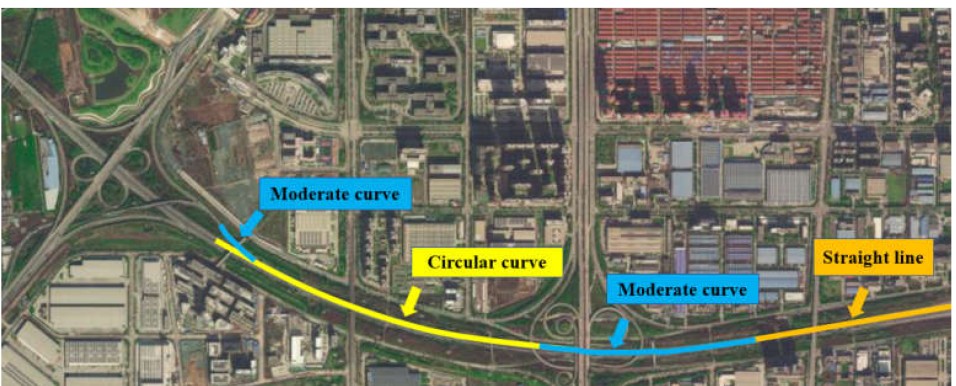

**Figure 6.** Geometric alignment of the simulation scene.

Based on this, 36 exit models are shown in Figure 7 and the real effect in the driving simulator when participants drove on a right curve with a radius of 1245 is shown in Figure 8.

After driving in one scenario, the participants completed a questionnaire that contained one question, "Do you feel comfortable driving under this scenario". There were the following five options to choose from, A: Very comfortable, B: General comfortable, C: Just acceptable, D: A little uncomfortable, E: Very uncomfortable.

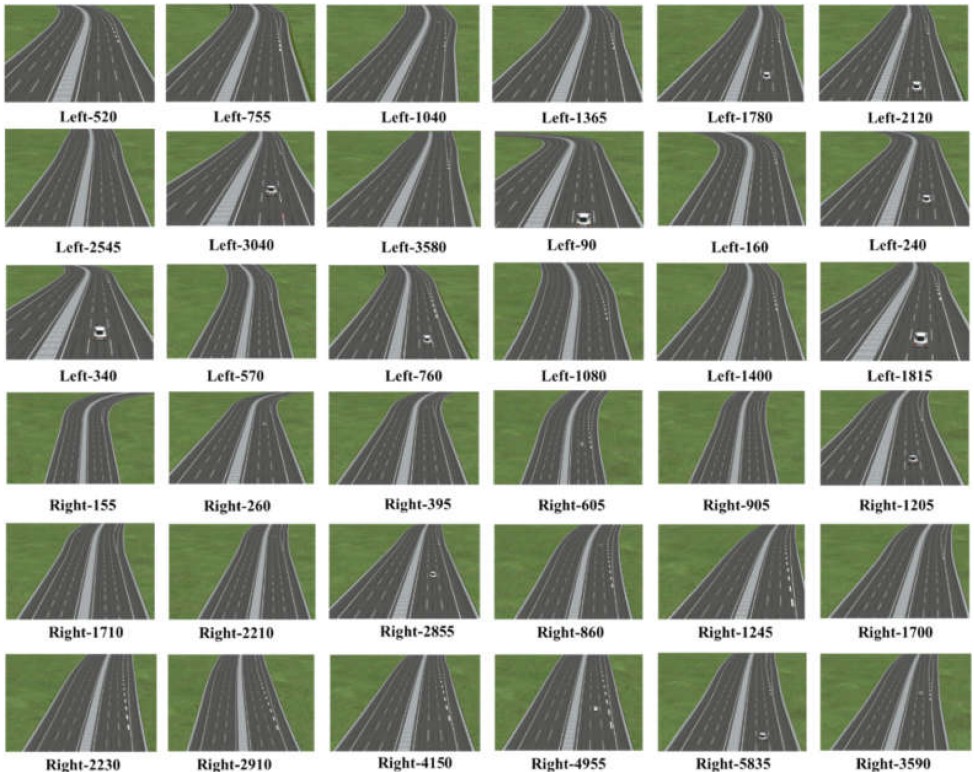

**Figure 7.** Radius setting in thirty-six scenarios.

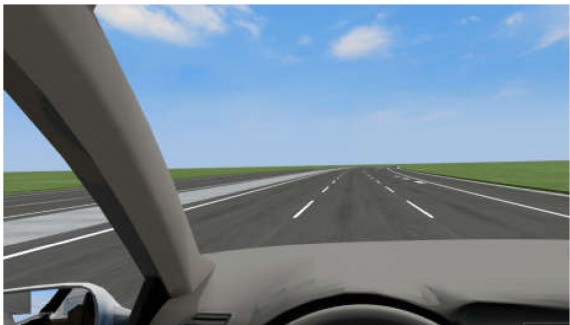

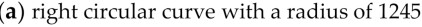

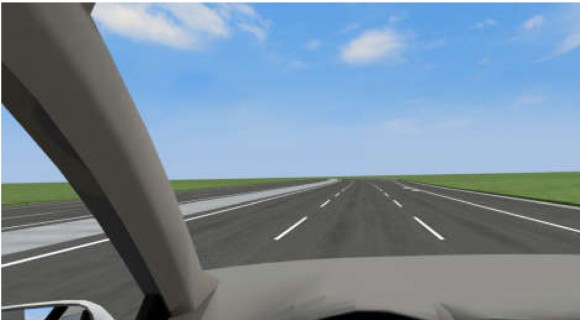

(**a**) right circular curve with a radius of 1245 | (**b**) left circular curve with a radius of 1245

**Figure 8.** Simulation effect in the simulator. The left picture (**a**) is the simulation effect shown on the middle liquid crystal display when participants drove in the scenario when the main line is a right circular curve with a radius of 1245 and the right picture (**b**) is a left circular curve with a radius of 1245.

### 3.5. Procedure

Since each participant had to drive in 36 scenarios (which took about two hours), to save time and avoid driving fatigue, they were divided into six groups, each with five participants. The participants took turns completing a scenario until all of the scenarios were completed. The sequence of scenes was random to avoid trials since the occurrence of each scene was random and 6 people completed the single scene that appeared in turn. The driver did not know the driving speed and the deflection of the circular curve in advance, the occurrence time of the exit ramp in each scene is inconsistent and the interval driving is realized, which can effectively prevent the driver from self-learning.

Before the experiment began, participants were asked to adjust their bodies to a comfortable sitting position. They were then told about the alignment of the road and tested in a simulator for five minutes to familiarize themselves with the operating system. In addition, the participants were told to drive as usual and at the specific speed limit for

each scenario. During the experiment, participants were required to drive in the middle lane until they had identified the exit from the freeway and were able to leave the main line. When the participants completed each scenario, they were asked to complete a questionnaire.

## 4. Analysis and Results

### 4.1. Thermodynamic Chart

After the end of the experiment, a total of 30 questionnaire results were collected and the results of each questionnaire were sorted into a thermodynamic chart, as shown in Figure 9. A thermodynamic chart is a kind of information expression that can reflect the distribution, density and trend of change in the object. Thermodynamic charts are widely used in data analysis and display. Some scholars use heat maps to analyze the motion rules of objects [47–49], while others use heat maps to show the selected frequency of objects. Figure 9 shows the percentage of drivers who chose different comfort levels in different scenarios in the 30 questionnaires. The comparison table at the top of the Figure shows the corresponding proportions of different colors. The "ABCDE" in the Figure represent the comfort level of the driver mentioned above.

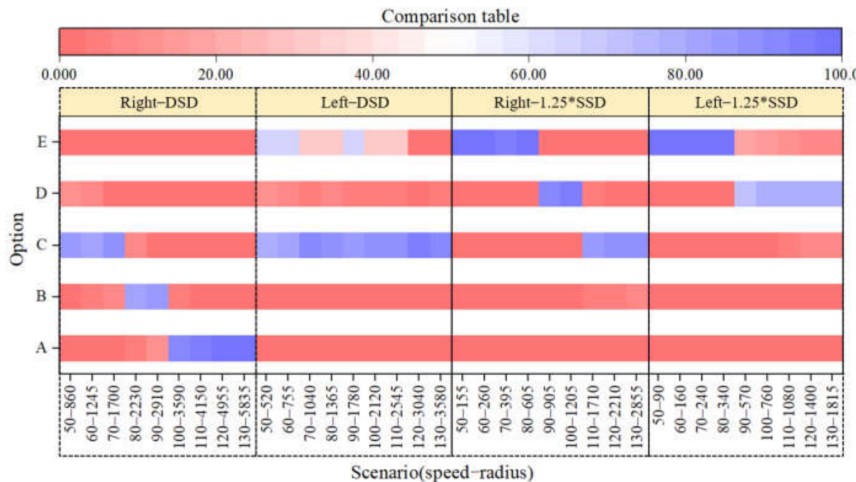

**Figure 9.** Questionnaire results. The "Right–DSD" in the figure indicates that this kind of scene is a right circular curve, and the radius of the circular curve is calculated based on the DSD.

As can be seen from Figure 9, in terms of the overall trend, drivers feel more comfortable when driving on a circular curve that satisfies the DSD, regardless of the circular curve's deflection. When the driver drives at the same speed on the right circular curve satisfying the DSD and the 1.25 times the SSD requirements, it is obviously more comfortable to drive on the decisive one. At the same time, it can be seen that when driving on circles with radii of 155, 260, 395 and 605 m, satisfying the 1.25 times the SSD requirement, most of the drivers indicated that they were "very uncomfortable" in their feedback, it is after the radius increases to around 1710 m that drivers can barely accept the experience of driving. On the contrary, when driving on the limit radius circle curve satisfying the DSD, the feedback of the driver in the worst case is "acceptable".

The situation is a little different between the left circular curve and the right circular curve. When the circular curve deviates to the left, the driver's driving experience feedback is mostly "barely acceptable", even when the driver is driving on a circular curve that satisfies the DSD. The driver feels terrible when driving on a circular curve that satisfies only the 1.25 times the SSD requirement. After asking most of the drivers, it is concluded that they need to change lanes to the right when driving on a left curve, which is a driving behavior contradiction. Therefore, we suggest that the main line geometric design in a diverging area should be avoided as far as possible by selecting a left circular curve.

### 4.2. Steering Wheel Angle Rate and Steering Wheel Angle Frequency Domain

The steering wheel angle rate (SAR) reflects the operating load of the driver [50–53]. The SAR refers to the change of the steering wheel angle (SWA) in unit time. The greater the SAR is, the faster the driver is turning the steering wheel. At this time, the more likely a vehicle is to skid or roll over and crash, thus affecting traffic safety.

Currently, there are several ways to judge lane changes. It can be judged by detecting lane line changes [54–57], SWA changes [58,59] and vehicle drift angle changes [60]. In this paper, the vehicle trajectory is derived using the Forum 8 Driving Simulator. By corresponding to the alignment file number of the road, the trajectory data of vehicles in the diverging area when they change lanes is intercepted. Thus, data on the SWA and the SAR during the operation period can be found. For example, the change of the SWA and the SAR in the process of lane change is shown in Figures 10 and 11. It can be seen from Figure 11 that the SWA and SAR have no obvious changes at the beginning of the segment and then SAR suddenly increases, which stipulates that the driver starts to change lanes when the SWA suddenly increases. Point "S" is determined as the initial time of a lane change. When the lateral velocity is zero, the SWA and the SAR tend to be zero, and this is defined as the end time of a lane change, that is point "E" in Figure 11. By using this method, the starting and ending time of a lane change in 36 cases is uniformly defined in this paper to ensure the consistency of the research problem.

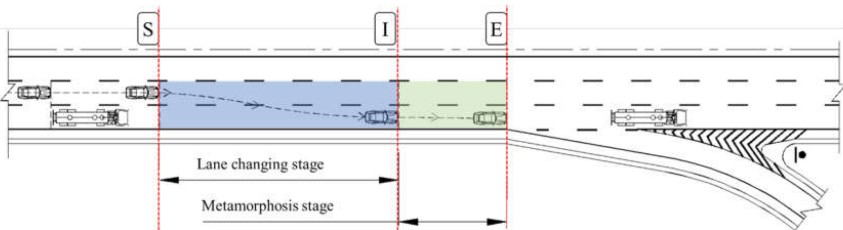

**Figure 10.** Lane changing process.

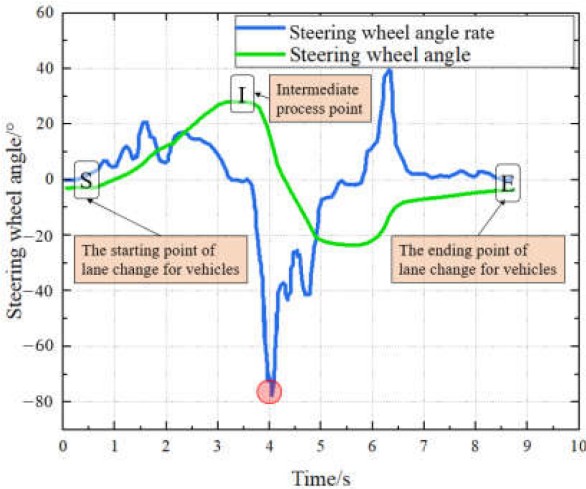

**Figure 11.** Driving data display in the process of a vehicle lane change.

In order to increase the accuracy of the conclusion, this paper uses Fourier Transform to transform the SWA in the time domain into the frequency domain, to analyze the information of the SWA in the frequency domain and the driver's operating busyness during the whole lane changing process. The frequency domain diagram obtained is shown in Figure 12.

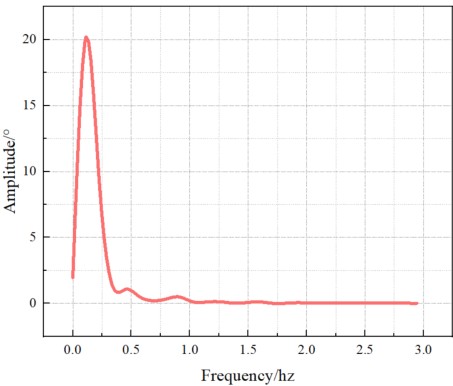

**Figure 12.** Steering wheel angle frequency domain.

The steering wheel angle frequency domain (SWAF) reflects the driver's operating response. When the SWA is more than 15° and at a frequency of less than 0.4 Hz, the driver's operating response is relatively slow, which can be considered to mean that the driver is not busy with the operation. The higher the frequency is, the busier the driver is. A moderate SAR and SWA can make the driver more comfortable in the process of driving and make the workload intensity smaller, which is conducive to driving safety. Many studies have realized that the driver's SAR and SWA [61] are important factors to ensure driving safety [62,63] and these two indicators are also affected by other factors. In this study, except for the size of the radius of the circular curve, other environmental conditions are the same; therefore, we can take SAR and SWAF as important measurement indexes.

The SWA and SAR data of each driver in the 36 scenes were extracted using the Forum8 Driving Simulator. The SWA and SAR of the 30 participants in the 36 scenes were tested using the normality test, as shown in Table 3. The distribution of these data was normal, and it has mathematical statistical significance.

The origin is used to plot the SAR of the driver's steering wheel and the result is shown in Figure 13.

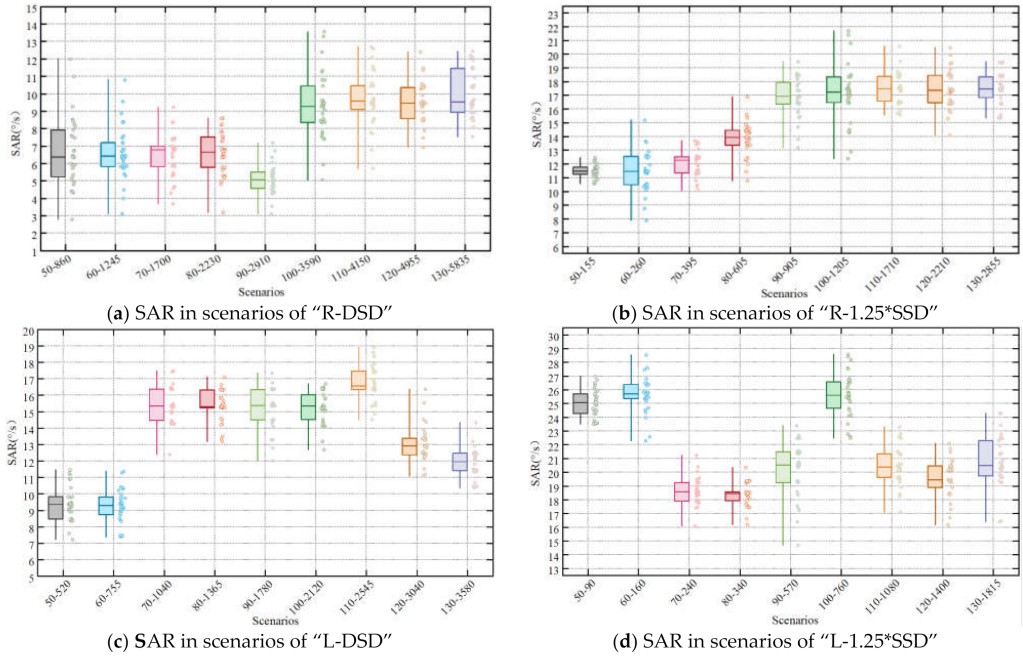

**Figure 13.** SAR of the drivers. In the abscissa 50–860 (e.g., **a,b**) represents the driving speed (km/h)—the radius of the circle curve (m).

**Table 3.** Normality test of the SAR and SWA.

| Scenarios | Speed/(km/h)-radius/s | Mean | Statistical | df | Sig. | Mean | Statistical | df | Sig. |
|---|---|---|---|---|---|---|---|---|---|
| | | | SAR | | | | SWA | | |
| Right circular curve that meets DSD (R-DSD) | 50–860 | 6.675 | 0.961 | 30 | 0.336 | 20.364 | 0.967 | 30 | 0.466 |
| | 60–1245 | 6.620 | 0.957 | 30 | 0.264 | 19.522 | 0.944 | 30 | 0.122 |
| | 70–1700 | 6.472 | 0.969 | 30 | 0.522 | 20.634 | 0.978 | 30 | 0.787 |
| | 80–2230 | 6.585 | 0.968 | 30 | 0.490 | 20.742 | 0.953 | 30 | 0.206 |
| | 90–2910 | 5.093 | 0.974 | 30 | 0.661 | 18.526 | 0.961 | 30 | 0.332 |
| | 100–3590 | 9.285 | 0.965 | 30 | 0.433 | 20.964 | 0.963 | 30 | 0.378 |
| | 110–4150 | 9.663 | 0.954 | 30 | 0.223 | 25.268 | 0.977 | 30 | 0.752 |
| | 120–4955 | 9.482 | 0.973 | 30 | 0.626 | 25.662 | 0.955 | 30 | 0.231 |
| | 130–5835 | 9.991 | 0.946 | 30 | 0.135 | 25.759 | 0.965 | 30 | 0.417 |
| Right circular curve that meets 1.25 times SSD (R-1.25*SSD) | 50–155 | 11.510 | 0.979 | 30 | 0.816 | 21.747 | 0.986 | 30 | 0.957 |
| | 60–260 | 11.486 | 0.964 | 30 | 0.407 | 22.407 | 0.965 | 30 | 0.431 |
| | 70–395 | 12.029 | 0.960 | 30 | 0.319 | 22.462 | 0.935 | 30 | 0.067 |
| | 80–605 | 13.824 | 0.961 | 30 | 0.346 | 34.339 | 0.951 | 30 | 0.189 |
| | 90–905 | 16.904 | 0.955 | 30 | 0.233 | 37.686 | 0.934 | 30 | 0.066 |
| | 100–1205 | 17.147 | 0.941 | 30 | 0.101 | 37.166 | 0.959 | 30 | 0.296 |
| | 110–1710 | 17.485 | 0.948 | 30 | 0.172 | 37.396 | 0.979 | 30 | 0.816 |
| | 120–2210 | 17.422 | 0.972 | 30 | 0.604 | 37.133 | 0.969 | 30 | 0.534 |
| | 130–2855 | 17.447 | 0.936 | 30 | 0.071 | 36.977 | 0.959 | 30 | 0.305 |
| Left circular curve that meets DSD (L-DSD) | 50–520 | 9.380 | 0.952 | 30 | 0.192 | 20.401 | 0.981 | 30 | 0.860 |
| | 60–755 | 9.260 | 0.961 | 30 | 0.331 | 21.373 | 0.959 | 30 | 0.294 |
| | 70–1040 | 15.369 | 0.940 | 30 | 0.092 | 30.571 | 0.935 | 30 | 0.068 |
| | 80–1365 | 15.346 | 0.933 | 30 | 0.059 | 28.390 | 0.969 | 30 | 0.523 |
| | 90–1780 | 15.200 | 0.932 | 30 | 0.058 | 37.096 | 0.970 | 30 | 0.552 |
| | 100–2120 | 15.250 | 0.937 | 30 | 0.079 | 33.396 | 0.974 | 30 | 0.672 |
| | 110–2545 | 16.769 | 0.969 | 30 | 0.521 | 30.943 | 0.973 | 30 | 0.645 |
| | 120–3040 | 13.060 | 0.933 | 30 | 0.061 | 28.610 | 0.961 | 30 | 0.345 |
| | 130–3580 | 11.907 | 0.940 | 30 | 0.094 | 28.658 | 0.939 | 30 | 0.086 |
| Left circular curve that meets 1.25 times SSD (L-1.25*SSD) | 50–90 | 25.029 | 0.959 | 30 | 0.300 | 46.312 | 0.968 | 30 | 0.497 |
| | 60–160 | 25.624 | 0.935 | 30 | 0.067 | 46.256 | 0.962 | 30 | 0.361 |
| | 70–240 | 18.636 | 0.960 | 30 | 0.317 | 40.467 | 0.970 | 30 | 0.562 |
| | 80–340 | 18.381 | 0.942 | 30 | 0.108 | 38.369 | 0.963 | 30 | 0.378 |
| | 90–570 | 20.106 | 0.943 | 30 | 0.111 | 40.357 | 0.983 | 30 | 0.901 |
| | 100–760 | 25.642 | 0.965 | 30 | 0.424 | 45.571 | 0.973 | 30 | 0.650 |
| | 110–1080 | 20.492 | 0.974 | 30 | 0.667 | 36.865 | 0.966 | 30 | 0.438 |
| | 120–1400 | 19.541 | 0.960 | 30 | 0.321 | 36.123 | 0.939 | 30 | 0.088 |
| | 130–1815 | 20.822 | 0.943 | 30 | 0.111 | 36.591 | 0.952 | 30 | 0.191 |

The driver's operating load can be judged using the SAR. It can be seen from Figure 13, that when the driver is driving in the "R-DSD" scene, the overall SAR is lower than $10°/s$ and the driver's operating load is small. Moreover, the driver's operating load in this scene is the lowest among them all. However, when the driver is driving in the "L-1.25*SSD" scene, the overall operating load is the highest and the SAR is between 18 and $30°/s$, indicating that the driver's operating load is high, which easily causes the driver to be too nervous and increases the difficulty of vehicle control, thus leading to operational errors and even traffic crashes.

The frequency band of 0~0.4 Hz is the driver's inactive frequency band. The proportion of inactive frequency band (PIFB) in the total frequency band is used to observe the driver's driving condition. Then, the driving condition of the driver in each scene can be obtained through statistics, as shown in Figure 14.

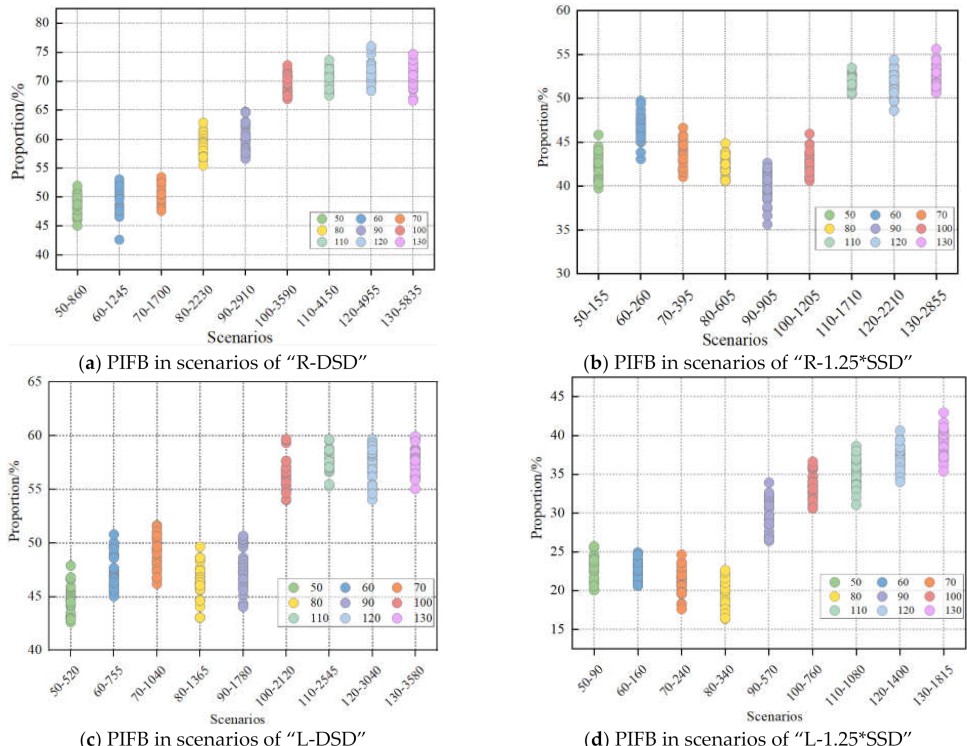

(**a**) PIFB in scenarios of "R-DSD"

(**b**) PIFB in scenarios of "R-1.25*SSD"

(**c**) PIFB in scenarios of "L-DSD"

(**d**) PIFB in scenarios of "L-1.25*SSD"

**Figure 14.** The PIFB in each scenario.

Figure 14 shows the PIFB when changing lanes in different scenes. In different scenes, the trend of changing is different. It shows that the difference in speed, radius and circular curve deflection affect the driving experience of the driver. In the "R-DSD" scene, the PIFB increases slowly at first with the increase in speed. In the scene corresponding to "70–1700"~"100–3590" km/h, the PIFB increases at a faster rate and then tends to be stable. Moreover, in the "R-DSD" scene, the PIFB of the driver is between 45 and 75%, indicating that the driver operates smoothly and does not panic when driving in this kind of scene. In the "R-1.25*SSD" scenario, the PIFB is between 35 and 55%. In the "L-DSD" scene, the driver's PIFB is between 40 and 60%. In the "L-1.25*SSD" scene, the PIFB of the driver is between 15 and 45% and the driver is almost in a stage of panic operation for a long time, which can easily cause operational errors and traffic crashes.

At the same time, it can be seen from Figures 13 and 14 that when the vehicle's speed is certain, the operation of the driver is relatively stable when driving on a road that meets the DSD, the SAR is low and the PIFB is large. However, when driving on a road that meets the 1.25 times the SSD requirement, especially on the left circular curve, the SAR is large, the PIFB is small, the driver's operating load is large and driving is nervous, which can easily cause operating errors and even traffic crashes.

## 5. Discussion

The main line in a diverging area that meets the need of DSD can significantly affect a driver's operating load. On a road that meets the DSD, a driver can drive more smoothly and feel more comfortable, reduce the SAR and increase the PIFB, which is conducive to the safety of a driver in a diverging area, as shown in Figure 15.

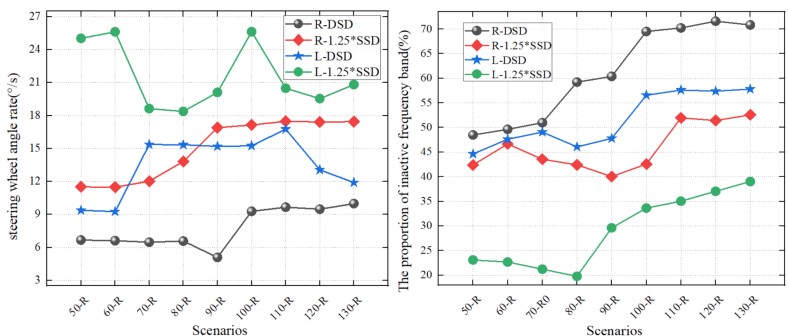

**Figure 15.** The SAR and PIFB in scenarios (on the x-axis, similar "50-R" and "60-R" represent the driving speed and the radius of the circle curve in different scenes, which is also called the "speed-radius").

At the same time, in order to further judge the influence of a left/right circle curve on drivers' driving performance, statistical difference analysis was conducted on the SAR and the PIFB of driving on the left/right circle curves satisfying DSD (Table 4).

**Table 4.** Mean and SD of the SAR and the PIFB when driving on a left/right circle curve with the same DSD.

| Deviation | | R-DSD | | | | L-DSD | | | | ANOVA | |
|---|---|---|---|---|---|---|---|---|---|---|---|
| | Index | SAR(°/s) | | PIFB(%) | | SAR(°/s) | | PIFB(%) | | SAR | PIFB |
| Speed/(km/h) | | Mean | SD | Mean | SD | Mean | SD | Mean | SD | | |
| 50 | | 6.675 | 2.000 | 48.484 | 1.654 | 9.380 | 1.041 | 44.634 | 1.378 | F = 43.177 $p \leq 1.534 \times 10^{-8}$ | F = 9 5.968 $p \leq 6.616 \times 10^{-14}$ |
| 60 | | 6.620 | 1.555 | 49.627 | 2.314 | 9.260 | 1.025 | 47.618 | 1.559 | F = 60.218 $p \leq 1.547 \times 10^{-10}$ | F = 15.537 $p \leq 2.202 \times 10^{-4}$ |
| 70 | | 6.472 | 1.227 | 50.983 | 1.512 | 15.369 | 1.084 | 49.061 | 1.603 | F = 885.514 $p \leq 8.005 \times 10^{-37}$ | F = 22.818 $p \leq 1.252 \times 10^{-5}$ |
| 80 | | 6.585 | 1.241 | 59.217 | 1.797 | 15.346 | 0.967 | 46.072 | 1.644 | F = 930.143 $p \leq 2.092 \times 10^{-37}$ | F = 873.385 $p \leq 1.165 \times 10^{-36}$ |
| 90 | | 5.093 | 0.847 | 60.383 | 2.315 | 15.200 | 1.237 | 47.790 | 1.780 | F = 1363.148 $p \leq 5.494 \times 10^{-42}$ | F = 557.646 $p \leq 1.940 \times 10^{-31}$ |
| 100 | | 9.285 | 1.909 | 69.518 | 1.614 | 15.250 | 0.988 | 56.558 | 1.453 | F = 230.884 $p \leq 6.976 \times 10^{-22}$ | F = 1067.441 $p \leq 4.791 \times 10^{-39}$ |
| 110 | | 9.663 | 1.568 | 70.233 | 1.650 | 16.769 | 1.112 | 57.604 | 1.136 | F = 409.692 $p \leq 5.709 \times 10^{-28}$ | F = 1190.855 $p \leq 2.338 \times 10^{-40}$ |
| 120 | | 9.482 | 1.182 | 71.605 | 1.845 | 13.060 | 1.130 | 57.336 | 1.427 | F = 143.575 $p \leq 2.511 \times 10^{-17}$ | F = 1122.293 $p \leq 1.203 \times 10^{-39}$ |
| 130 | | 9.991 | 1.353 | 70.845 | 2.074 | 11.907 | 0.868 | 57.806 | 1.306 | F = 42.556 $p \leq 1.841 \times 10^{-8}$ | F = 848.414 $p \leq 2.565 \times 10^{-36}$ |

At the exit of the diverging area, the DSD and the 1.25 times the SSD have different influences on the driving comfort, the difficulty of vehicle handling and the psychological tension of the driver. The driving effect of a right circular curve is different from that of a left circular curve. Under the same conditions, driving on a right circular curve is more favorable for the driver to maneuver the car safely. When driving on a circular curve that satisfies the DSD, even if the speed is 130 km/h, most drivers still feel comfortable. However, most drivers feel very uncomfortable when driving on a circular curve that meets the 1.25 times the SSD requirement.

On the other hand, when driving on a right circular curve that satisfies the DSD, the driver's SAR is the lowest when changing lanes. At the same time, the PIFB during the lane change phase is also the highest. It shows that the driver's maneuvering difficulty is low, and the maneuvering does not panic the driver, which is conducive to driving safely. The worst case is driving on a left circular curve that satisfies the 1.25 times the SSD requirement. In this scenario, the driver's SAR is very large and PIFB is very low. In this situation, the driver is nervous in driving, has difficulty manipulating the vehicle and traffic crashes, such as vehicle sideslip, are prone to occur.

Furthermore, when the deflection is the same, the driving feeling, driving difficulty and driving load of a driver on an exit meeting the DSD are much better than those that meet the 1.25 times the SSD requirement, and the driving is more comfortable and safer. At the exit of the diverging area, the main line adopts a right circular curve, which is more conducive to safe driving.

Statistical difference analysis (Table 4) showed that the left/right circle curves were different for drivers' driving performance. Under the condition of the same DSD, the mean and SD of the SAR and the PIFB are better when driving on a right circle curve, indicating that the driver is more free and more leisurely. In actual situations, they should also treat this differently. At present, it may be a security hazard to treat the sight distance requirements of left and right circular curves uniformly. Finally, taking 1.25 times the SSD in a diverging area may also aggravate the occurrence of traffic crashes.

## 6. Conclusions

Through a simulator study, 30 participants' driving data were collected. By analyzing these data, this paper explored the impact of 1.25 times the SSD and DSD at the exit of a freeway diverging area on driving safety. The 30 participants' subjective evaluation of the driving comfort were obtained using a survey questionnaire. The conclusions are as follows:

- When driving at the exit of a diverging area that satisfies the DSD, the driver's manipulation difficulty and driving load can be reduced. It can help the driver better leave the main line.
- When driving on a circular curve that satisfies the DSD, regardless of the speed, the SAR of the vehicle is the lowest when changing lanes and the PIFB accounts for the largest proportion. In terms of driving safety, it is recommended that the main line at the exit of a diverging area needs to meet the DSD. Compared to a left circular curve, the driving index on a right circular curve is more ideal. The left circular curve does not meet driving expectations.
- According to the subjective questionnaire, most participants stated that it was more comfortable to drive at the exit of a freeway diverging area that meets the DSD. Moreover, almost all of the participants thought that driving on a right circular curve was more comfortable than driving on a left circular curve. The subjective evaluation further confirms the conclusion drawn from the simulation experiment data.
- It may be problematic to treat the minimum radius and sight distance requirements of left and right circular curves equally. Taking 1.25 times the SSD in the diverging area may also aggravate the occurrence of traffic crashes.
- In summary, this paper discusses the impact of 1.25 times the SSD and the DSD at the exit of a freeway diverging area on driving safety. The results indicate that it is more favorable for a driver to operate calmly and ensure the safety of driving at an exit that satisfies the DSD. However, our experimental design did not consider the influence of the longitudinal alignment of the road on the driving of the driver. In future research, we will improve this aspect of the design.

**Author Contributions:** Conceptualization, X.Z. and B.P.; data curation, X.Z.; formal analysis, X.Z.; methodology, X.Z. and Y.S.; software, X.Z.; visualization, X.Z. and B.P.; writing—original draft, X.Z.; writing—review and editing, B.P. All authors have read and agreed to the published version of the manuscript.

**Funding:** This research received no external funding.

**Institutional Review Board Statement:** The study was conducted according to the guidelines of the Declaration of Helsinki and approved by the Institutional Review Board of School of Highway, Chang'an University.

**Informed Consent Statement:** Informed consent was obtained from all subjects involved in the study.

**Data Availability Statement:** The DOI for the data is 10.6084/m9.figshare.14721627.

**Conflicts of Interest:** The authors declare no conflict of interest.

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
