# Peer review of "Evaluating the Impact of Sight Distance and Geometric Alignment on Driver Performance in Freeway Exits Diverging Area Based on Simulated Driving Data"

_sustainability, doi:10.3390/su13116368_

Round 1
Reviewer 1 Report
Review comments on "Evaluating the Impact of Sight Distance and Geometric Alignment in Freeway Exit Diverging Area Based on Simulated Driving Data."
This study aimed to analyze the impact of 1.25 times of stopping sight distance and decision sight distance on drivers' performance at the exit of the diverging freeway area. To achieve the purpose, this study used a driving simulator to develop 36 scenarios. The steering wheel angle rate, steering wheel angle frequency domain, and the questionnaire results were collected for final analysis. Overall the idea of the paper is pretty good, and the results are well presented. However, few points still need to be improved regarding the experiment design, results, and technical writing.
The introduction part was combined with literature review which is fine. But if the focus is on the DSD, SSD, and circular curve deflection, the authors need to categorize the “literature review” to these topics. It would be better if the authors start a paragraph with the topic, list literature review results, and end with the summary and need for further study (which is relevant to this paper). The literature review in the paper is confusing to me. It is great that in the last paragraph, the authors summarized the literature review results and pointed out the purpose of this paper. However, there are no summarized objectives in the introduction. It would be helpful if the authors can list them in bullets.
As for the participant recruitment for the experiment. How did the authors ensure that 30 people is enough sample size? What is the age distribution for those participants? Did the age distribution of participants follow the same trends of the driver age distribution in china? Based on the literature I reviewed for the driving simulator study, most of them will provide the mean and standard deviation of participants' age. It is suggested to provide a standard deviation of the age for this study. Regarding the 36 similar scenarios created for this study, how did the authors ensure that the driver did not self-learning while driving?
It is surprising to see the different impacts of the left/right circular curve on the driver's performance. It seems like the authors applied the statistical analysis to figure out whether the data collected in the study was normal and has statistical significance. It is also recommended to apply the statistical analysis on the data to future proof the impacts of the left/right circular curve on the driver's performance is a statistical difference.
As for the technical writing, it is highly recommended the authors do the proofreading. There are lots of grammar errors, incomplete sentences, and typos in the manuscript. Some words are capitalized in the sentence for no reason. Three important things that the authors need to aware of: i) I think the author did not use "lane" and "line" properly in the paper (e.g.: "main line" in pg2, line 51); ii) the DSD and SSD can be used in the whole paper except the first time use; and iii) It is unusual to use "accident" in the technical paper, the authors should use "crash," instead.
Other minor issues can be found as follow:
- Page1, line 3: add "on driver performance" between alignment and in.
- Page1, line 18: this sentence is too long and did not make any scene. Try to break it down.
- Page1, line 26: change "road system" to "freeway system."
- Page1, line 30-32: do you use any reference on this sentence? You may need to provide a specific year and freeway name in this sentence.
- Page1, line 38: change "a drivers's driving safety" to "affect driver safety".
- Page 1 line 11, change “Based on the field road environment of Hechizai interchange…” to “based on Hechizai Interchange…”.” the field road environment of” seems redundant.
- Page 1 line 13, insert a comma between “(SSD)”and “and…”.
- Page 1 line 15, change “subjects” to “volunteers”.
- Page 1 line 17, change the sentence to “the overloaded driving tasks and operation would be more likely to cause crashes”.
- Page 1 line 18 to 19, I do not understand this sentence. I guess it is saying “…is better than that on the left circular curve, because the lane change….”
- Page 1 line 19 to 21, if this sentence is the conclusion, try to use bullet. For example, other findings are summarized as follows: (1) ;(2); and (3).
- Page 1 line 26, change “accident” to “crash” for the rest of the paper.
- Page 1 line 39, change “… of bridge” to “the bridge”.
- Page 1 line 42, change to “…by better infrastructure or traffic flow…”.
- Page 1 line 44, change the semicolon to period.
- Page 2 line 54, change to “this was not paid much attention to”.
- Page 2 line 58, change “lines” to “pavement markings”.
- Page 2 line 59, delete “relevant” before “control”.
- Page 2 line 60, delete “driving” or change to “when driving at the exit ramp…”
- Page 2 line 61, “obscured”, are you referring to “obstructed”?
- Page 2 line 60 to 64, “when…line”, modify this sentence to multiple ones.
- Page 2 line 63, change “drive away” to “diverge from”.
- Page 2 line 68 to 69, I do not understand the sentence “of which 45…ramp exit”. It would be better for the authors to divide the long sentence into short ones.
- Page 2 line 88, change “…meet the …” to “…meeting…”.
- Pg 2, Line 49: change "there are also" to "The other".
- Pg2, line 49: change "accident rate" to "crash rate".
- Pg2, line 60-64: the sentence is too long to read. Please break it down.
- Pg2, line 68: change "statistical period" to "study period".
- Pg2, Line 70: change "And" to "In addition".
- Pg2, line 77-80: grammar error.
- Pg2, line 83-86: grammar error
- Page 3 line 99, change “decision sight distance…” to “DSD”.
- Page 3 line 111, delete “driving”.
- Page 3 line 121, delete the second “that”. Same in page 4 line 146.
- Page 3 line 124, ”in a timely…”?, same in Page 4 line 149.
- Page 3 Figure 2, SP is the driver’s view point position, but in Figure 1, view point is at the TP location. This is confusing, please modify. Same with page 4 Figure 3. There is explanation towards the term “viewpoint” on page 5. But seems this view point on page 5 is different from which in figure 1. Please explain.
- Page 4 line 145 to 154, please modify your wording. 2.2.1 and 2.2.2 are exactly the same. If they are the same, the authors do not need to address twice.
- Page 5 line 180 to 182, please edit the sentence, grammatical errors.
- Page 7 line 220,” three-lane road each direction”?
- Page 7 line 223 to 224, change “PGDHS” to “the Green Book”.
- Page 10 line 280, change to “…feedbacks are…”.
- Page 10 line 285, change to “a little different”.
- Page 15 line 409, change to “is very large”.
- Page 15 line 407 to 408, change “safe driving of the driver” to “drive safely”.
- Pg 6, line 209: did you mean "15,000km" and "180,000km"?
- Pg 7, line 223: change "by" to "based on the method in the."
- Pg 10, line 294: change "Steering" to "steering."
- Pg 10, line 300: explain what is UC-Winroad
- Pg10, line 305: change "Angle" to "angle."
- Pg 15, Figure 15: need to double-check the notations on the x-axis. I think it supposes to be the speed limits. I don't think it is left/right curve-related.
Reviewer 2 Report
Improve the wording and coherence of the text, for example:
- Delete ")" in line 78
- sentence in 87-88
- figure 2 in line 176 should be figure 4
- value "b" in line 182 should be parameter D1
- Comment on dates of the shooting process with UAV in 177
- Variable "D" from formula (3) in 188 could be Ds from formulae (1) and (2)?
- Line data 200 does not correspond to that shown in Table 1 (188)
- Name the variables in formulae (1) and (2) in the text of lines 220-222
- Correct "radii" in 229
- separate radii number in line 279
- Change "." to "," in line 280
- Improve column headings in table 3.
Improve the explanation of the process followed and some of the results:
- how is D2 obtained from formula (3) in 188)?
- How are the vehicle types differentiated in Table 1 (192)?
- For the calculation of the radius indicated on line 224, how do you calculate S1 and S2 from DSD and/or 1.25*SSD?
- Add image of left circular curve with radius 1245 to complement Figure 8.
- For a better understanding of Figure 9, it is proposed to use a maximum of 5 or 7 ranges, e.g. with 15% or 20% bands instead of 10% bands. Or use two shades of colour, to better differentiate between close ranges.
- Explanation of the value 100-760 in Figure 13d, line 346.
Formatting of references 53 (lines 565-567) and 55 (lines 570-572)
Reference cited documents:
- Linhai North Interchange of Shenhai Expressway in China (65-70)
- Forum8 Driving Simulator
It is recommended to review works by Bassani and Gargoum, for example:
Gargoum, S.A., Tawfeek, M.H., El-Basyouny, K., Koch, J.C. 57189216298;57217582094;14832701400;57200183473; Available sight distance on existing highways: Meeting stopping sight distance requirements of an aging population (2018) Accident Analysis and Prevention, 112, pp. 56-68. Cited 13 times. https://www.scopus.com/inward/record.uri?eid=2-s2.0-85040072486&doi=10.1016%2fj.aap.2018.01.001&partnerID=40&md5=241ce25fb86db2a808f366250c06a83d DOI: 10.1016/j.aap.2018.01.001 DOCUMENT TYPE: Conference Paper PUBLICATION STAGE: Final Bassani, M., Catani, L., Salussolia, A., Yang, C.Y.D. 26532382800;57190399659;57209749636;36613901700; A driving simulation study to examine the impact of available sight distance on driver behavior along rural highways (2019) Accident Analysis and Prevention, 131, pp. 200-212. Cited 3 times. https://www.scopus.com/inward/record.uri?eid=2-s2.0-85068592996&doi=10.1016%2fj.aap.2019.07.003&partnerID=40&md5=88675439ba544f62b10753eaae6341c3 DOI: 10.1016/j.aap.2019.07.003 DOCUMENT TYPE: Article PUBLICATION STAGE: Final Bassani, M., Hazoor, A., Catani, L. 26532382800;57211072249;57190399659; What's around the curve? A driving simulation experiment on compensatory strategies for safe driving along horizontal curves with sight limitations (2019) Transportation Research Part F: Traffic Psychology and Behaviour, 66, pp. 273-291. Cited 2 times. https://www.scopus.com/inward/record.uri?eid=2-s2.0-85072559133&doi=10.1016%2fj.trf.2019.09.011&partnerID=40&md5=c843568d85560259f7510dfb1588b2e9 DOI: 10.1016/j.trf.2019.09.011 DOCUMENT TYPE: Article PUBLICATION STAGE: Final Gargoum, S.A., El-Basyouny, K. 57189216298;14832701400; Analyzing the ability of crash-prone highways to handle stochastically modelled driver demand for stopping sight distance (2020) Accident Analysis and Prevention, 136, art. no. 105395, . Cited 1 time. https://www.scopus.com/inward/record.uri?eid=2-s2.0-85076704409&doi=10.1016%2fj.aap.2019.105395&partnerID=40&md5=003d4e52391a3da63e96606b772d099a DOI: 10.1016/j.aap.2019.105395 DOCUMENT TYPE: Article PUBLICATION STAGE: FinalAuthor Response
Please see the attachment
